# Polymer Nanocomposite Membrane for Wastewater Treatment: A Critical Review

**DOI:** 10.3390/polym14091732

**Published:** 2022-04-24

**Authors:** Sivasubramani Divya, Tae Hwan Oh

**Affiliations:** School of Chemical Engineering, Yeungnam University, Gyeongsan 712-749, Korea

**Keywords:** polymer nanocomposite membrane, nanofiller, fouling, permeability

## Abstract

With regard to global concerns, such as water scarcity and aquatic pollution from industries and domestic activities, membrane-based filtration for wastewater treatment has shown promising results in terms of water purification. Filtration by polymeric membranes is highly efficient in separating contaminants; however, such membranes have limited applications. Nanocomposite membranes, which are formed by adding nanofillers to polymeric membrane matrices, can enhance the filtration process. Considerable attention has been given to nanofillers, which include carbon-based nanoparticles and metal/metal oxide nanoparticles. In this review, we first examined the current status of membrane technologies for water filtration, polymeric nanocomposite membranes, and their applications. Additionally, we highlight the challenges faced in water treatment in developing countries.

## 1. Introduction

Water is the most precious resource for living organisms on Earth. In recent years, water contamination has become a global environmental concern. The primary reason is that population growth, industrialization, and climate change affect drinking water resources, such as rivers and lakes [1,2,3,4]. According to the World Health Organization (WHO), unsafe drinking water, sanitation, and hygiene are responsible for 10 million deaths annually, mainly caused by infectious diarrhea [5]. To address this challenge, it is important to reclaim water resources using wastewater treatment and water separation technologies. In this context, several methods have been developed to serve different wastewater technologies, such as conventional filtration, coagulation-flocculation, and biological treatment [6].

Membrane technology holds great potential for wastewater treatment because of its small size, low energy consumption, and low initial cost. A membrane is a barrier that selectively allows desired materials to pass through, with undesired materials retained on the membrane surface [6], such as polymeric and inorganic membranes. Metal- or ceramic-based inorganic membranes provide high structural, mechanical, and thermal resistance. Although they are highly selective, their low permeability makes them unsuitable for a variety of applications. Contrarily, polymer membranes present good flexibility, chemical stability, mechanical strength, easy fabrication, and are inexpensive materials. The chemical species were selectively transferred. Materials used for the fabrication of polymeric membranes include polyvinyl alcohol (PVA), polyether sulfone (PES), polyvinylidene fluoride (PVDF), polyvinyl chloride (PVC), polypropylene (PP), polyacrylonitrile (PAN), polyamide (PI), polyethylene (PE), polyamide (PA), and chitosan [7,8,9]. 

Several studies have investigated wastewater treatment using polymer membrane technology. Nasreen et al., 2013 [10], studied PVDF and hydroxyethyl methacrylate (HEMA) membranes with improved flux and antifouling properties for micro water filtration. Li et al. [11] designed PVA and polypropylene (PP) electrospun membranes for microfiltration. The electrospun membrane exhibited a permeate flux of 32,346 L/m^2^ h and a pressure of 0.24 bar. Elele et al. (2019) [12] observed microfiltration membranes of PES and PVDF and tested the mechanical properties of stress, strain, amplitude oscillation, and bubble point pressure. Ong et al., 2021 [13], discussed a polycarbonate membrane with enhanced solvent resistance for microfiltration applications. Yan et al. [14] studied the use of PVDF for ultrafiltration membranes and achieved improved surface roughness, permeability, and antifouling properties. Orooji et al. [15] investigated tetraphenylethylene with polyether sulfone for wastewater purification via ultrafiltration. The superior luminescence properties of composite membranes make them an ideal choice for non-destructive and biofouling monitoring. Johari et al. [16] studied PES blended with an iron-based metal-organic framework for ultrafiltration membranes, which was suitable for industrial wastewater treatment; this hybrid membrane showed remarkable rejection and high permeation flux above 98.5% and 165.68 L/m^2^ h. Li et al. [17] examined nanofiltration membranes for fouling and chemical cleaning using water quality control. Zhang et al. [18] studied polyamide membrane for nanofiltration applications. A maximum selectivity of 246 for sodium sulfate with sodium chloride, and 51 for magnesium chloride with sodium chloride, indicated its excellent ability for solute-solute separation. Tang et al. [19] investigated the removal of toxic substances from wastewater mixed with perfluorooctane sulfonate (PFOS) using reverse osmosis (RO) and nanofiltration (NF) processes. The removal efficiency of RO membranes was above 99% and that of nanofiltration was up to 99%. Tain et al. [20] studied the treatment of brackish groundwater using a mixture of RO and reverse osmosis technologies. Their results addressed flux, salt rejection, and energy consumption. Elazhar et al., 2021 [21], removed and reduced the chloride and fouling rejection rates through the NF-RO system. 

Overall, this review article discusses a comprehensive collection of polymer nanocomposite membranes and their application in wastewater treatment. Moreover, our review also focuses on water treatment in different countries.

## 2. Polymer Nanocomposite Membrane

Blend- and film-based nanocomposites are two types of polymer nanocomposite membranes. Nanoparticles-encapsulated membranes or nanoparticles-mixed membranes are terms used to refer to nanocomposite membranes. Nanoparticles and polymers are dispersed in casting solutions in blended nanocomposite membranes, and their applications are listed in Table 1. Behboudi et al. [22] prepared a blend membrane of PVC with PC using a non-solvent-induced phase separation method. The pore radii of the blended membranes ranged from 2 to 4 nm. The wettability test of the blend membranes confirmed the hydrophilic nature of the membranes, and the water flux of the 70 wt.% PC membrane was 1260 kg/m^2^ h (Figure 1). In addition, abrasion and stability tests confirmed the stability and elongation of the blend membranes. Furthermore, the antifouling properties of the membranes were obtained using fouling parameters, such as the total fouling ratio, reversible fouling ratio, irreversible fouling ratio, and flux recovery [23]. Chen et al. [24] studied biocomposite polymers made with rice husk fiber by melt extrusion or compression molding, which was also treated with gamma radiation (25–150 kGy). The effect of gamma radiation changed the tensile properties at different weight loadings. After irradiation, an increase in the modulus and a decrease in the elongation at the break of the composite membrane were observed (Figure 2).

In recent years, polymer thin-film nanocomposite membranes have attracted attention for water purification because of their hydrophilicity, thermal stability, selectivity, permeability, and thermal resistance. Generally, nanofillers-based thin-film composite membranes are metal-, metal oxide-, and carbon-based materials, and their applications are listed in Table 2. Zhao et al. [35] synthesized a zeolitic imidazole framework-8 (ZIF-8) incorporated in a polymer nanocomposite membrane for the desalination of brackish and seawater. In terms of water permeation, imperfections in the ZIF-8 could prevent water molecules from transporting resistance while they pass through the nanoparticles (pictorial representation in Figure 3a). The separation performance with respect to the time is shown in Figure 3b. The water flux value of the composite membrane is higher than that of the pure thin-film composite membrane (TFN). Therefore, TFN membranes containing dZIF-8 can be used for long-term RO desalination without significantly degrading their performance. Liu et al. [36] substituted Zr into a polyamide membrane for water desalination applications. Ramokgopa et al. [37] investigated the permeate flux and rejection ratio of carbon nanotubes in polymer nanocomposite films, which aid the removal of heavy metal ions during acid drainage.

Rafiei et al. [38] studied the mechanical, thermal, and fouling properties of PVDF nanocomposite films via polymerization reactions. The figure shows that the addition of metal oxide to PVDF increases the flux rate and improves the surface roughness and thermal stability [39,40]. Lim et al. [41] studied the application of polyethylene terephthalate nanocomposite films in the food industry and discussed the structural, water vapor permeability, and oxygen permeability of the films. The results showed that the addition of montmorillonite improved the barrier properties of the film (Figure 4). Therefore, PET nanocomposite films can potentially be used in food packaging with good gas barrier abilities for oxygen-sensitive foods [42,43,44].

## 3. Application of Polymer Nanocomposite Membrane 

Application for wastewater management is classified into two types: (1) qualitative data and (2) quantitative data, such as flow diagrams, water consumption, water contaminants, and effective solutions for wastewater treatment. This section reviews water usage in industries and proposes water-consumable management, sustainable production, and environmental protection. Figure 5 shows a schematic of the application of the water treatment.

### 3.1. Automotive Industry

With the development of modern science and technology, the automobile industry has become an important sector that enhances productivity and high-quality vehicles to fulfill people’s requirements. However, hazardous compounds are produced during the manufacturing process, which has detrimental effects on the environment and human health. Additionally, this sector consumes a large amount of water (nearly 182,000 L) [57,59,60]. The automobile manufacturing process is a complex process, as shown in Figure 6. High-quality water is needed for all stages in the manufacturing process as it is used to mix chemicals and cleaners, and must fulfill the manufacturer’s specifications to ensure that the product quality does not suffer. The following stages of the automotive manufacturing process consume the most water, as listed in Table 3.

The recycling of water has been working in some countries; for example, Belgium consumes 0.3% of the several million liters used widely in the carwash industry; Germany and Austria utilize 80% of the water produced by the recycling process, and European countries consume an average of 70 L per car [61]. Researchers have focused on water consumption and wastewater management in the automobile industry using physical, chemical, and biological methods, such as filtration, coagulation, flocculation, ozonation, and aerobic and anaerobic processes. Table 4 summarizes the methods and removal efficiency in the automobile industry.

### 3.2. Food Industry

The meat processing industry is one of the largest industries with an annual production of 690 million chickens in the commercial boiler sector [68]. The global production of chicken and sheep meat is expected to expand in lockstep with animal numbers, although the output of pig meat, beef, veal, and milk will grow at a faster rate than the animal population. In particular, poultry output in China and Latin America is expected to reach nearly 40% of worldwide poultry meat expansion. In Europe, the rate of increase in poultry meat production per animal has decreased in recent years, and production is likely to remain stable in the following years. By 2028, beef and veal production is predicted to increase by 9 MT. Latin America and the United States, the world’s two largest producing regions, will account for more than half of the worldwide increase of such outputs [69]. The meat industry uses a large quantity of water for cleaning and processing purposes, and a schematic representation of meat processing is shown in Figure 7. In the meat industry, the lairage, slaughter, and bleeding sections, as well as the dressing area, paunch handling area, rendering unit, and processing and cleaning sections collectively produce waste. In slaughterhouse plants, water consumption reaches 2.5–40 MT of meat produced. A large percentage of the consumed water is discharged as wastewater that is severely polluted with chemical oxygen demand (COD), biological oxygen demand (BOD), total phosphorous (TP), total nitrogen (TN), and total organic carbon (TOC) [70]. Table 5 shows wastewater treatment in the meat production industry. There are numerous methods to preserve water during the meat manufacturing process, including: (1) physicochemical treatment, such as dissolved air flotation, coagulation, flocculation, and electrocoagulation, which are efficient in removing total suspended solids (TSS); (2) for slaughter wastewater treatment, membrane technology is another method used for slaughter wastewater treatment; RO, NF, UF, and MF processes remove particles, colloids, and macromolecules depending on the membrane pore size; and finally, (3) biological treatment and aerobic and anerobic methods for the removal of soluble organic compounds. 

### 3.3. Beverage Industry

In the twentieth century, a major revolution was observed in brewery industries due to increased consumer demand. The beverage industry is divided into two categories: (1) alcoholic (wine and beer) and (2) non-alcoholic (fruit juices, tea, coffee, carbonated soft drinks, and bottled water). Both types of beverages are widely consumed worldwide. America dominates the world’s soft drink market with the next leading countries being China, Mexico, Asia, Mexico, and Brazil [78,79]. A large amount of water is used in beverage industries for rinsing, soaking, cleaning, and washing, and nearly half of the wastewater is discharged [80]. Hence, this has led to the depletion of natural water resources. 

Tea is one of the most consumed beverages in the world. Turkey, Ireland, the United Kingdom, Russia, and Morocco are the world’s largest tea drinkers, each consuming over 10 kg of tea per capita annually [81]. Tea manufacturing processes vary depending on the type of tea being produced, such as green, yellow, oolong, black, and dark tea. A schematic representation of the six types of tea production is shown in Figure 8. The water pollutants from tea factories include turbidity, COD, BOD, TSS, TOC, and pH. Table 6 summarizes recent research work focused on wastewater treatment.

### 3.4. Pharmaceutical Industry

According to the World Health Organization (WHO), medical waste is generated from hospitals, research, and laboratory centers. Such wastes include drugs, paper wipes, gloves, masks, syringes, needles, bandages, and dressing cloths (Figure 9). Inadequate disposal of infectious hospital waste and wastewater can endanger public health and the environment [91,92]. The rapidly growing population is the primary factor driving the increasing usage of medical products in the developed world, and higher usage is causing a corresponding increase in medical waste production. America alone produces nearly 3.5 million tons of medical waste per year [93,94,95]. Pharmaceutical wastewater contains BOD, COD, organic contaminants, nitrogen, and suspended solids. In the pharmaceutical manufacturing industry, water is an important requirement for container sterilization, medical devices, and injections. Wastewater is generated by different pharmaceutical units, including manufacturing, extraction, processing, purification, and packaging. However, pharmaceutical waste is not the only contaminant in the environment. Aquatic and terrestrial species are also exposed to pesticides, biocides, and waste from chemicals industries [96]. Table 7 lists the pharmaceutical wastewater treatments reported in the literature.

### 3.5. Wastewater Treatment

Contamination of water bodies occurs due to waste from industrial sectors, such as organic pollutants (food, dyes, pesticides, herbicides, detergents, and pharmaceuticals), inorganic pollutants (heavy metal ions and rare earth elements), and other pollutants (oil, spill, grease, radioactive waste, etc.) [106,107,108]. As shown in Figure 10, different technologies are used to remove contaminants present in wastewater. These techniques are efficient for removing effluent from wastewater, while membrane-based techniques separate the floating and dispersed solids, exhibiting high efficiency and low energy requirements [109,110]. Membrane bioreactors are sustainable water and wastewater treatment technologies that unite biological reactors with filtration to overcome the disadvantages of traditional membrane systems. However, membrane fouling is the major limitation that prevents them from being used on a larger basis in the application of bioreactors [111]. Antifouling or self-cleaning or novel materials are one of the most common techniques for dealing with membrane fouling. Recent studies on polymer nanocomposite membranes have focused on enhancing their antifouling properties. Shahkaramipour studied the antifouling performance of zwitterions modified with polydopamine and sulfobetaine methacrylate using an ultrafiltration membrane. The results showed that the flux efficiency was higher than that of an uncoated membrane [112]. Incorporation of a hydrophobic epoxy polymer into a hydrophilic poly(N-isopropylacrylamide) polymer coated with silver nanoparticles enhances the antifouling properties [113]. Moreover, the coated samples were tested in a marine field for 45 days (Figure 11A), and the temperature, oxygen, and pH level of seawater were maintained (Figure 11B). Iron oxide nanoparticles were added to poly(2-dimethylamino)ethyl methacrylate and meso-2, 3-dimercaptosuccinic acid with a poly(methacrylic acid) nanocomposite membrane by the phase separation method. The results indicated that hydrophobic and hydrophilic membranes enhanced the permeate flux by applying a magnetic field and fouling properties without a magnetic field [114]. Phin et al. [115] fabricated zinc oxide and carbon nanotubes coated with polysulfone nanocomposite membranes using the phase-inversion method. The results showed that the antifouling strength improved with the addition of nanoparticles. The fouling ratio (Rir) indicates the irreversibility of foulants that are strongly attached to the membrane surface and inside the membrane pore. Heidari et al. [116] synthesized polyethersulfone (PES), sulfonated PES (SPES), and phosphotungstic acid with cesium hydrogen salt (CsPW/SPES) of a hybrid membrane using a phase-inversion method for nanofiltration applications. The results revealed a high filtrate volume of CsPW/SPES of 0.035 m^3^/m^2^ compared to pure PES and SPES. The permeability rate increased with the addition of nanoparticles compared to that of pure PES. Hydrophilicity influences the membrane permeation rate through the contact between water molecules and membrane pore walls. Castro et al. [117] coated graphene oxide on a polymer using a phase-inversion method for the removal of phytopathogenic fungi in water. Zhang et al. [118] reported the removal of dye with the inclusion of graphene oxide in polyacrylonitrile by interfacial polymerization (Figure 11C). The results show that the water flux of the nanocomposite membrane increases at 0.2 MPa in the range of 15–23 L/m^2^h greater than the bare membrane, and dye rejection was achieved [118].

## 4. Water Treatment in Different Countries

### 4.1. Technology Used for Wastewater Treatment

Due to their high population, many developing countries are struggling to provide clean water to their populations. With limited government resources, ensuring that all areas in developing countries have access to clean water is a difficult undertaking [119,120]. To meet their daily freshwater demand, rural villagers in economically developing countries, such as Kenya, have turned to alternative rainwater harvesting systems [121]. However, in South Africa, borehole water has been found to have significant levels of nitrate-nitrogen and pH, making it unsafe to drink [122]. A conventional water treatment system is still cost-effective, but water resources are heavily contaminated by industrial wastewater [123]. Other dangerous dissolved contaminants could not be eliminated using these techniques. Membrane filtration systems have higher removal efficiencies, particularly when UF and RO membranes are combined. For instance, membrane systems have even been utilized in developed countries, such as Singapore, to recycle sewage water into high-grade treated water from industrial wastewater [124] and for the past few decades, China’s rapid industrialization has proposed research and development (R&D) on massive UF membrane water treatment plants [125]. Detailed information on the water treatment in different countries is listed in Table 8. To date, commercial-level membrane technology has been used to produce drinkable water from brackish water or seawater. Pretreatment methods are recommended in most studies to ensure that the feed water remains suitable for the membrane system. This method helps to improve membrane efficiency and life duration by reducing fouling.

### 4.2. Reuse of Wastewater in Irrigation

Generally, all effluents of wastewater are from households, institutions, hospitals, industries, and other sources, such as surface runoff, urban runoff, agriculture, horticulture, aquaculture, and cattle breeding. Pathogens, heavy metals, residual medications, organic chemicals, pharmaceuticals, and health care items are among the contaminants found in many types of wastewater [140,141,142]. Therefore, it is important to treat wastewater before use. Different technologies, techniques, and preparations for wastewater treatment are discussed in the previous section. Treated wastewater (TWW) has been used for agricultural, industrial, and alternative activities in the developed world. Below are some examples of how other countries have used treated wastewater for irrigation. Tunisia has recently begun to use TWW for irrigation. Mahjoub et al. discussed reusing TWW in the agricultural sector in Tunisia [143]. In 1965, wastewater was used to irrigate 1200 ha of orchards. In the years from 1965 to 1989, the reuse of TWW in the agriculture sector was supported by government sectors [144,145]. During this period, intense research has been conducted on the reuse of TWW in the agricultural sector as a result of its environmental impact. By 2020, half of the wastewater had been targeted for irrigation. In many regions of Lebanon, most wastewater is used for irrigation. The domestic and industrial sectors produce approximately 310 million m^3^ of wastewater, of which approximately 4 million m^3^ was processed and used to irrigate farm fields in 2006 [146]. The massive increase in Kuwait’s population, along with increasing industrialization and water demand for agriculture, has led to a sharp decline in surface and groundwater. Considering the water shortage, the Kuwait government developed more than one wastewater treatment plant that produces 76 gallons of treated effluents per day [147]. Some Syrian cities, such as Damascus, Aleppo, Homs, Hama, and Salamiyeh, have treated wastewater used for irrigation using enhanced surface irrigation systems [148]. Due to natural and technological limitations, Palestine (the West Bank and Gaza Strip) is one of the Middle East’s most water-scarce countries. Poor sanitation, insufficient wastewater treatment, unsafe disposal of untreated or partially treated water, and the use of untreated wastewater to irrigate edible crops characterize the wastewater sector in the West Bank and Gaza (WBG). The occupied Palestinian territory (OPT) now contains eight main urban WWTPs, with around 300 on-site treatment units [149]. Previously, wastewater and effluent were dumped in rural areas, river beds, and small towns, as well as in the Arabian Sea. To meet water demand, the Saudi Arabia government has planned to treat 100% of wastewaters by 2025 [150,151]. Saudi Arabia has already surpassed the United States and China as the world’s third-largest water reuse market [152]. Riyadh has successfully irrigated nearly 9000 hectares of date palms and forage crops. In Oman, wastewater treatment includes more than 400 plants. Municipalities use most TWW for landscape irrigation. All wastewater networks and treatment plants across the country have received a USD 2.8 billion investment for creation, management, operation, and maintenance [153,154].

## 5. Conclusions and Outlook

This review focuses on the use of membrane technologies to address the challenges of water purification. Improvements in the properties of the membrane, such as hydrophilicity, thermal stability, selectivity, and permeability, have been achieved in recent decades, and may be further taken forward by fabricating polymer nanocomposite membranes. Even though impressive progress has been made in the filtration process, more research is still needed to overcome existing challenges. The inclusion of nanomaterials in the polymer matrix has been demonstrated to enhance the chemical, mechanical, and thermal properties. However, they may be the cause of some drawbacks, such as: (1) the water flow pathway may be blocked by the addition of nanomaterials; (2) defects and non-selective porosity may form as a result of inadequate interaction between the polymer and nanoparticles, lowering selectivity and decreasing the polymer chain packing; and (3) poor filler dispersion in the polymer matrix may increase the risk of nanoparticle aggregation and agglomeration on the membrane surface or inner membrane surface, resulting in poor separation performance. To overcome these problems, proper pore tuning to obtain adequate pore size distributions at the membrane surface and other factors should be optimized to improve the chemical stability, mechanical stability, and long-term stability. This study intends to demonstrate the resumption of industrial wastewater in developing countries. Wastewater treatment is a major concern in many countries because high levels of unwanted or unidentified pollution are extremely hazardous to humans and the environment. Countries such as Tunisia, Lebanon, Palestine, Kuwait, Syria, and Saudi Arabia use wastewater treatment in the agricultural sector. It helps ensure the long-term viability of water supplies, the environment, agriculture, and human existence around the world.

## Figures and Tables

**Figure 1 polymers-14-01732-f001:**
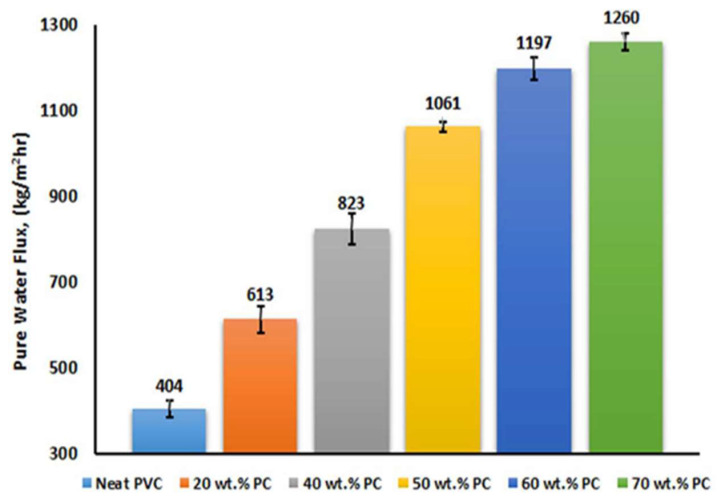
Water flux of polymer blend membranes (reproduced with permission from [22] copyright 2017, Elsevier).

**Figure 2 polymers-14-01732-f002:**
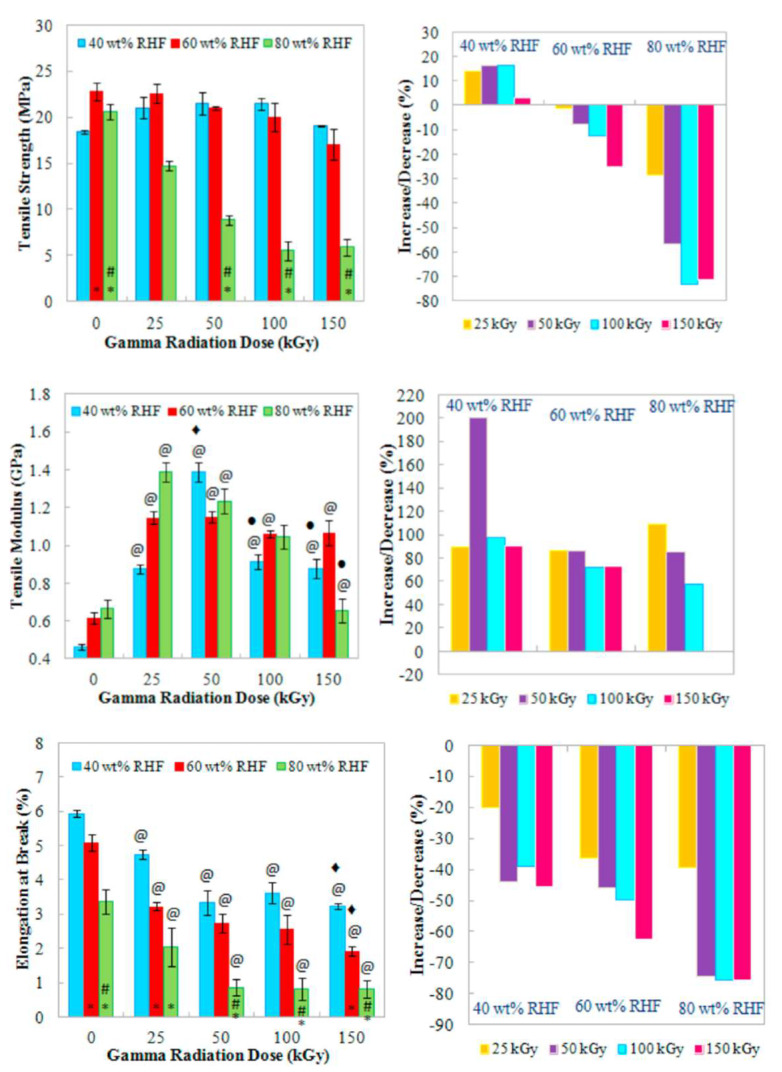
Tensile properties with applied gamma radiation. The symbols for gamma radiation * and # denotes 40 and 60 wt.% of rice husk fibers, for gamma radiation doses as compared to unirradiated, 25 and 50 kGy are are denoted by @, ◆ and ● (reproduced with permission from [24] copyright 2021, Elsevier).

**Figure 3 polymers-14-01732-f003:**
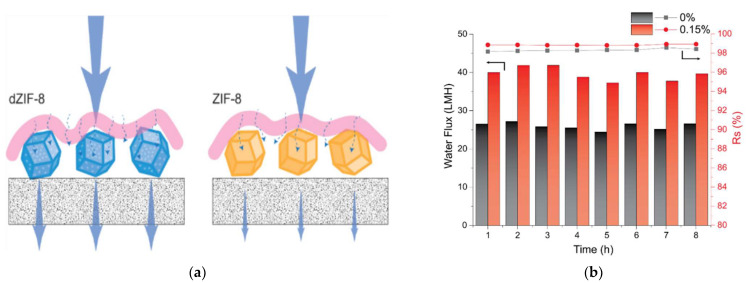
(**a**) Illustration of water permeance, (**b**) membrane desalination studies with dZIF-8 concentrations in seawater (reproduced with permission from [35] copyright 2021, Elsevier).

**Figure 4 polymers-14-01732-f004:**
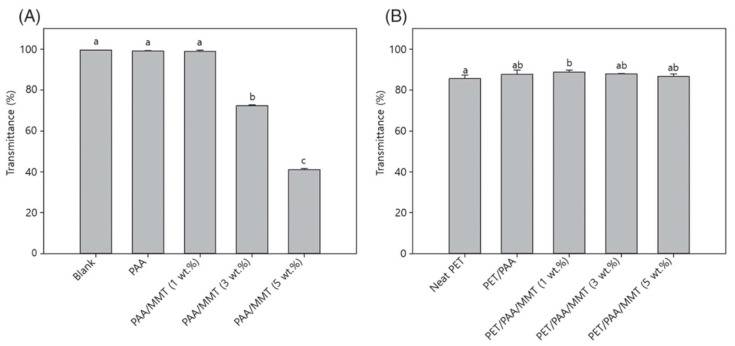
Barrier properties of water vapor (a) and oxygen permeability of polymer nanocomposite. (**A**) the polyacrylic acid (PAA)/montmorillonite (MMT) and (**B**) the polyethylene terephthalate (PET) films coated with PAA/MMT nanocomposites (reproduced with permission from [41] copyright 2021, Wiley).

**Figure 5 polymers-14-01732-f005:**
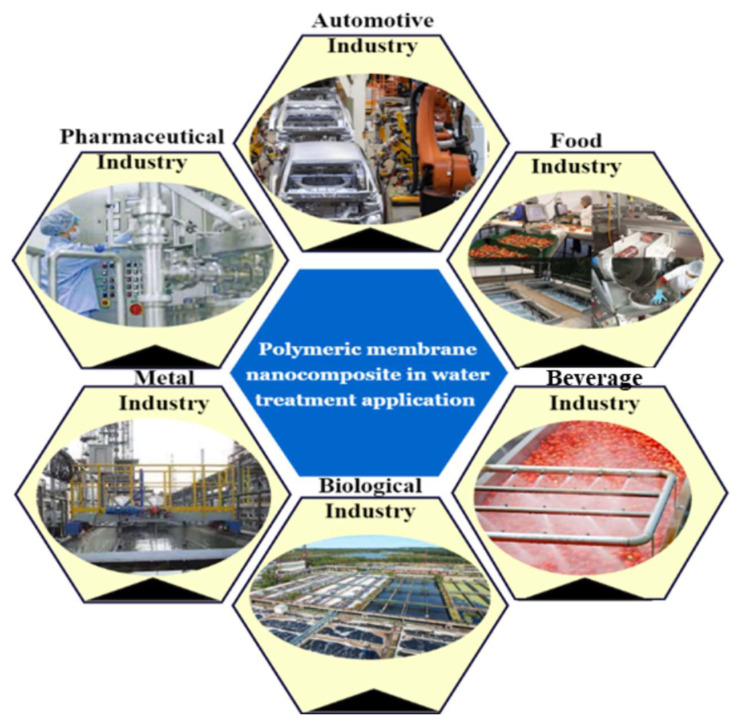
Schematic illustration of water treatment application.

**Figure 6 polymers-14-01732-f006:**
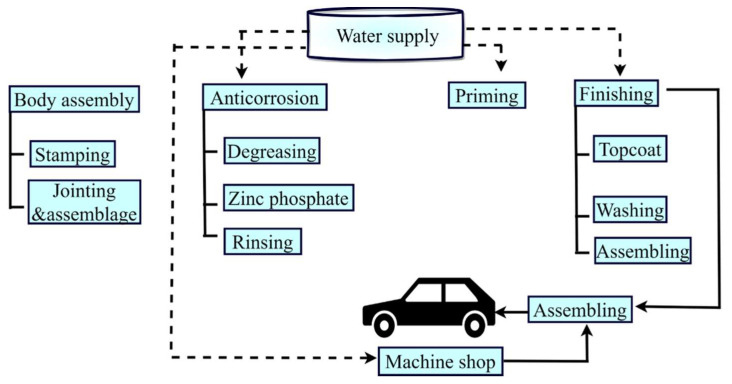
Representation of automobile production process.

**Figure 7 polymers-14-01732-f007:**
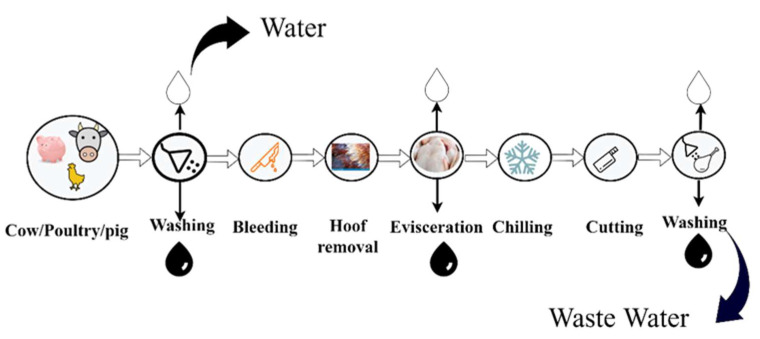
Schematic representation of the meat processing industry.

**Figure 8 polymers-14-01732-f008:**
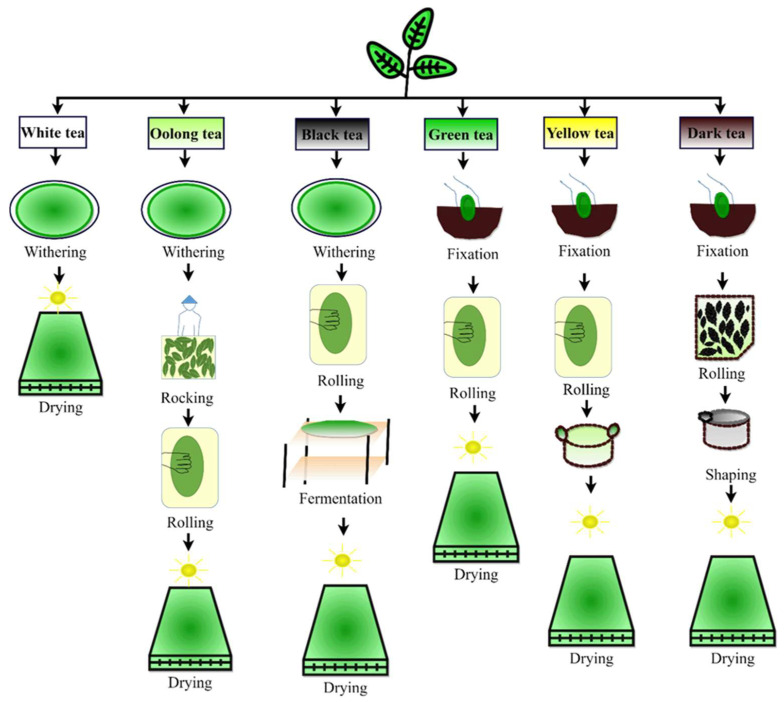
Tea manufacturing process.

**Figure 9 polymers-14-01732-f009:**
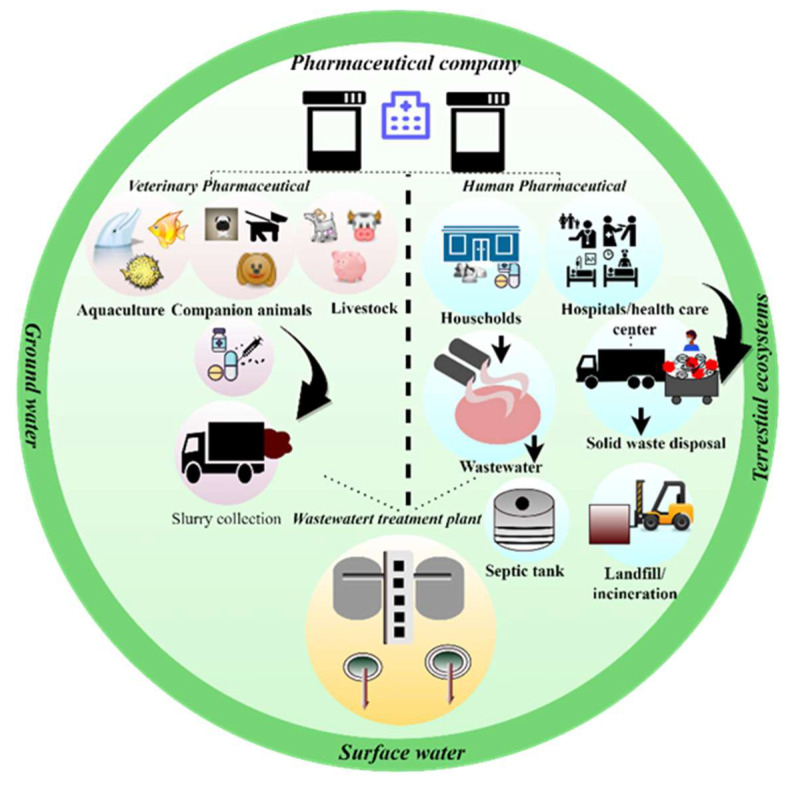
Pharmaceutical waste and wastewater disposal in the environment.

**Figure 10 polymers-14-01732-f010:**
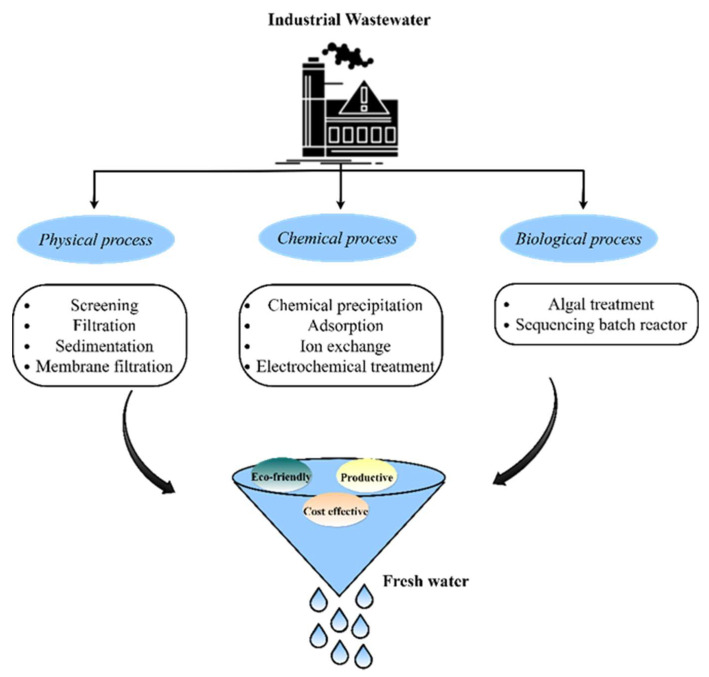
Industrial wastewater treatment.

**Figure 11 polymers-14-01732-f011:**
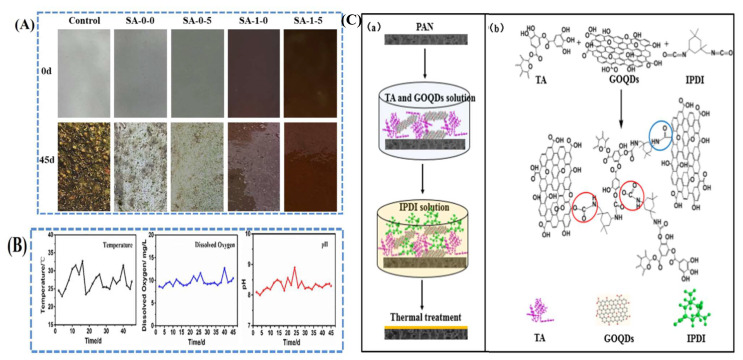
(**A**) Picture of samples submerged in seawater till 45 days; (**B**) temperature, oxygen, and pH level of seawater (reproduced with permission from [113] copyright 2021, Elsevier); (**C**) preparation (a) and mechanism (b) of graphene oxide nanocomposite membrane (reproduced with permission from [118] copyright 2021, ACS).

**Table 1 polymers-14-01732-t001:** Application of blended polymer nanocomposite.

Nanomaterials	Blended Polymers	Method	Application	Reference
Alumina	Polycarbonate/ Polyurethane	Phase separation	Ultra-filtration	[25]
Nitrogen doped porous graphene oxide	Polyethersulfone	Non-solvent-induced phase separation	Waste water treatment (removal of dyes)	[26]
Graphene oxide	Polyethersulfone	Phase separation	Waste water treatment (removal of industrial dyes)	[27]
Silica	Polylactic acid/Polybutylene/Polypropylene carbonate/Polyhydroxybutyrate	Phase separation	Waste water treatment (removal of oil)	[28]
Silver/Fe_3_O_4_	Poly(vinyl alcohol)/Poly(acrylic acid)/	Electrospinning	Waste water treatment	[29]
Graphene oxide-sodium alginate	Polyethersulfone/polyvinyl alcohol	Phase inversion induced by immersion precipitation	Waste water treatment	[30]
Ag-doped ZnO@Fe_3_O_4_/MWCNTs	Poly acrylic acid (PAA)-modified polyamide (PA)	Chemical co-precipitation	Pharmaceutical Waste water (removal of Amoxicillin)	[31]
Magnetic starch	Polyurethane/4,4-methylene diphenyl diisocyanate	In situ co-precipitation	Wastewater treatment	[32]
Nano diamond	PVDF/PVP	Phase inversion	Pharmaceutical wastewater treatment (removal of chemical oxygen demand)	[33]
Aluminum silicate	Chitosan/PVA	Electrospinning	Wastewater treatment (removal of dye)	[34]

**Table 2 polymers-14-01732-t002:** Application of thin-film polymer nanocomposite.

Nanomaterials	Thin Film Polymers	Method	Application	Reference
MgFe_2_O_4_ and ZnFe_2_O_4_	Polysulfone	Sol-gel	Wastewater treatment	[45]
Graphene oxide	Polyamide	Interfacial polymerization	Nanofiltration (removal of heavy metals)	[46]
Copper 1,4-benzenedicarboxylate	Polyamide	Layered diffusion-mediated	Forward osmosis	[47]
Chitosan-wrapped multiwalled carbon nanotubes	Polyethersulfone	Facile pouring	Nanofiltration (removal of dye)	[48]
MOF-801	Polyacrylonitrile	Solvothermal	Ultrafiltration (heavy metal removal)	[49]
Copper	Polyamide	Interfacial polymerization	Reverse osmosis	[50]
Zr-MOFs	Polyvinylpyrrolidone	Assisted in-situ growth	Wastewater treatment (removal of dye)	[51]
Graphene oxide/MoS_2_	Polyamide	Interfacial polymerization	Wastewater treatment	[52]
Titania nanotubes and magnetite oxide	Polysulfone	Coprecipitation	Forward osmosis (removal of heavy metals)	[53]
Silica	Polydopamine	Stober	Nanofiltration	[54]
Zwitterion-silver	Polyethersulfone	Non-solvent induced phase separation	Forward osmosis	[55]
TiO_2_	Polyamide/Polyethersulfone	Interfacial polymerization	Nanofiltration	[56]
TiO_2_	Polyamide	Biphasic solvothermal	Reverse osmosis	[57]
TiO_2_	Polyethersulfone/Polysulfone	Immersion precipitation	Wastewater treatment (removal of humic acid)	[58]

**Table 3 polymers-14-01732-t003:** Water usage in the manufacturing process.

Stage	Water Quantity (Liters)	Operation
Anticorrosion	60,000	Degreasing: Immersing the entire frame in a tub with detergent to remove any remaining oil, organic and inorganic chemicals from the stamping and jointing process. It is known as alkaline wash. Zinc phosphate: Frame is immersed in a chemical tub to cover a coating of zinc phosphate. Rinsing: To remove unwanted elements
Priming and Top coat	Large quantities	Water based paints requires large quantities of water to dilute the paint and
Finishing	Large quantities	After the topcoat, the frame will be washed again with water to remove any remaining paint and debris.
Machine shop	Large quantities	To remove grease

**Table 4 polymers-14-01732-t004:** Wastewater treatment in the automobile industry.

Method	Material	Operating Parameters	Removal Efficiency	Reference
Physical method-micro and ultrafiltration membrane	Polyetherimide	Flux, turbidity, organic and inorganic carbon	Flux—40 L/m^2^ h; turbidity—98%; organic—2.7 mg/L and inorganic carbon—35.4 mg/L	[62]
Chemical method-coagulation and ozonation	Polyethersulfone	pH, suspended solid, turbidity, chemical oxygen demand, phosphate	pH—8.5; SS—4200 µg/L; turbidity—1000 NTU; COD—433 mg/L; NH_4_^+^—2.2 mg/L;	[63]
Chemical method-electrocoagulation	Ti electrode	pH, chemical oxygen demand, suspended solids, chloride, oil grease	pH—8; COD—500 mg/L; SS—320 mg/L; chloride—70 mg/L; oil-grease—120 mg/L	[64]
Chemical method-electrocoagulation	Fe and Al-electrode	pH, chemical oxygen demand, suspended solid and oil grease	pH—8; COD—560 mg/L; oil-grease—125 mg/L; SS—2300 mg/L	[65]
Chemical method-electrocoagulation	Fe–Fe, Al–Fe, and Al–Al	chemical oxygen demand	COD—79%	[66]
Physical method-membrane filtration	Polyethersulfone/polyamide	pH, COD, total solids	pH—7; COD—314 mg/L; SS—1054 mg/L	[67]

**Table 5 polymers-14-01732-t005:** Wastewater treatment in meat production industries.

Method	Parameter	Removal Efficacy (%)	Reference
Physicochemical treatment	BOD COD	70.3% 80.3%	[71]
Aerobic treatment	COD BOD	2.8 kg/m^3^ 1.8 m^3^	[72]
Electrochemical coagulation	COD	92.91%	[73]
Isolation	COD	11.8%	[74]
Electrochemical coagulation	COD	93.22%	[75]
Dynamic membrane filtration	COD	14.5%	[76]
Anaerobic filter membrane bioreactor	COD BOD	97% 82%	[77]

**Table 6 polymers-14-01732-t006:** Research work focused on wastewater treatment.

Method	Material	Removal Efficacy (%)	Reference
Coagulation/flocculation	Ferric chloride and polyelectrolyte	97%—TSS;TP—99%; COD—91%	[82]
Coagulation/flocculation	Polyelectrolyte	78%—COD; 74%—TSS;75%—TP	[83]
Membrane bioreactor	Polyelectrolyte	95%—COD and TSS	[80]
Membrane ultrafiltration	Polyethersulfone	96—COD	[84]
Membrane filtration	Activated carbon	Phenol	[85]
Membrane ultrafiltration	Polyaluminum ferric chloride	99%—COD/TOC	[86]
Membrane nanofiltration	Polyamide	9%-lead	[87]
Co-pyrolysis	Tea waste	30%—cadmium	[88]
Membrane	Tea waste	7.2 mg/g cadmium	[89]
Chemical co-precipitation	Green tea extract	52%—fluoride	[90]

**Table 7 polymers-14-01732-t007:** Wastewater treatment in the pharmaceutical industry.

Treatment	Pharmaceutical Waste Water	Efficiency (%)	Reference
Membrane bioreactor	Chemical oxygen demand (COD), pH	COD—5789 (mg/L); pH—7	[97]
Pharmaceutical industrial Wastewater treatment	Dissolved organic carbon, total nitrogen (TN), chemical oxygen demand (COD), total phosphorous (TP)	Not reported	[98]
Anaerobic treatment	Sulfamethoxazole (SMX)	SMX—40 (mg/L)	[99]
Membrane bioreactor	Total chemical oxygen demand (TCOD)	COD—90%	[100]
Pharmaceutical industrial park wastewater treatment plant	pH, total suspended solids (TSS), chemical oxygen demand (COD), total nitrogen (TN), ammonia (NH_3_) and total phosphorus (TP)	pH—7; TSS—120 (mg/L); TN—84 (mg/L); COD—328 (mg/L); NH_3_— 37 (mg/L);	[101]
Membrane bioreactor	Total dissolved solid (TDS), total chemical oxygen demand (TCOD)	TDS—25,925 (mg/L); TCOD—20%	[102]
Solid-phase bioremediation	Erythromycin, sulfamethoxazole (SMX), tetracycline (TET)	ERY—1.2 (mg/L); SMX—11.5 (mg/L); TET—1.5 (mg/L)	[103]
Membrane bioreactor	Dissolved organic nitrogen (DON)	DON—68%;	[104]
Membrane bioreactor	Chemical oxygen demand, Etodolac	COD—90%; Etodolac—99%	[105]

**Table 8 polymers-14-01732-t008:** Water treatment usage in different countries.

Country	Water source	Water Treatment	Result	Reference
Australia	Ground water	UF and RO	Removal of TC, TOC	[126]
Brazil	Brackish water	RO-desalination	Removal of sulfate, total suspended solids and fluoride	[127]
China	Reservoir	Hollow fiber UF	Removal of metals, coliform bacteria	[128]
China	Songhuajiang river	Ultrafiltration	Removal of COD, DOC	[120]
Egypt	Black water	MF	Removal of TSS, BOD, COD	[129]
Germany	Industry	MBR	Removal of COD	[130]
India	Pesticide contaminated surface water	NF and RO	Removal of microbial content	[131]
Indonesia	Arsenic contaminated surface water	NF	Removal of TDS and arsenate ions	[132]
Malaysia	Surface water and ground water	UF	Removal of heavy metals such as chromium, cadmium, zinc, copper, nickel and lead	[133]
Netherland	Surface water	UF and MF	Removal of suspended solids, COD and bacteria	[134]
South Africa	Ground water	Gravity driven UF	Removal of *E.coli*	[135]
Sri Lanka	Ground water	NF	Removal of DOC, TDS	[136]
Thailand	Freshwater	NF	Removal of divalent cations, DOC, TDS	[137]
Turkey	Sea water	RO and NF-desalination	Removal of coliform	[138]
Vietnam	Sea water	Air gap membrane distillation	Removal of TDS, TOC, arsenic, mercury	[139]

## Data Availability

Not applicable.

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
