# Peer review of "Polymer Nanocomposite Membrane for Wastewater Treatment: A Critical Review"

_polymers, 2022, doi:10.3390/polym14091732_

Round 1

Reviewer 1 Report

All of the comments have been addressed adequately. I will recommend it for publication.

Reviewer 2 Report

The authors corrected the paper according to reviewers suggestions and it should be accepted in present form.

This manuscript is a resubmission of an earlier submission. The following is a list of the peer review reports and author responses from that submission.

Round 1

Reviewer 1 Report

polymers-1625690

This paper provides a short review on polymer Nanocomposite Membrane for Waste Water Treatment. The review is too general and not comprehensive and requires many improvements before being considered for publication. It does not pose significant merit for possible publication in the current form. Few comments below can be used by the authors to improve the manuscript quality.

  • The first sentence in the abstract is too jargonic. Please revise.
  • The first paragraph in the intro is too long. It actually carries different points that can be presented in a few paragraphs.
  • Please check the error in the reference manager.
  • Section 2 is too basic and not necessary in this review. Please omit or make it worthy to be maintained.
  • Section 3 requires a large summary tables to lay out the big picture of available research in this field.
  • Section 4: subsections on each application is required with more in-depth discussion.

Reviewer 2 Report

Greetings,

The article presents valuable information but there are a few changes I see necessary.

These are:

Line 49 and 50- the sentence needs to be reformulated.

Line 52, 131, 139, 155, 172, 185- there is an error, please remove and add the correct phrase

Line 145 please move on the previous page, right under the figure

 Figure 5- It is not readable. Some of the words can be read only if you turn your head

Line 194- Please move the title on the next page

Please read the article and fix all language issues.

For a review I would add more information.

Reviewer 3 Report

In this manuscript the authors give a short review about polymer nanocomposite membrane for waste water treatment. Although the study is well written it does not have much to say and it requires mayor revision before accepting the manuscript for publication. Following suggestions must be considered during the revision of the manuscript:

  1. In abstract, the conclusion is not good and should be rewritten
  2. I do not like the introduction at all, because it is very general and does not cover the topic. For example, there is no mention of the importance of using polymer nanocomposite membrane for waste water treatment.
  3. The discussion of the paper is weak and little scientific discussion has been presented to explain the main topic of manuscript. The authors should provide a conceptual focus on the discussions. In my opinion, membrane filtration should described in a few sentence while implementation with more details about polymer nanocomposite membranes is needed. I strongly recommended that you try to address this issue.
  4. Check the references.